# Glycogen Metabolism Supports Early Glycolytic Reprogramming and Activation in Dendritic Cells in Response to Both TLR and Syk-Dependent CLR Agonists

**DOI:** 10.3390/cells9030715

**Published:** 2020-03-14

**Authors:** Kylie D. Curtis, Portia R. Smith, Hannah W. Despres, Julia P. Snyder, Tyler C. Hogan, Princess D. Rodriguez, Eyal Amiel

**Affiliations:** 1Undergraduate student researcher or research employee, University of Vermont, Burlington, VT 05405, USA; Kylie.Curtis@uvm.edu (K.D.C.); Portia.Smith@uvm.edu (P.R.S.); Tyler.Hogan@uvm.edu (T.C.H.); 2Cellular, Molecular, and Biomedical Sciences Graduate Program, University of Vermont, Burlington, VT 05405, USA; Hannah.Despres@uvm.edu (H.W.D.); Julia.P.Snyder@uvm.edu (J.P.S.); Princess.Rodriguez@uvm.edu (P.D.R.); 3Department of Biomedical and Health Sciences, University of Vermont, Burlington, VT 05405, USA

**Keywords:** glycolysis, glycogen, glucose, dendritic cells, metabolism, innate immunity

## Abstract

Dendritic cells (DCs) increase their metabolic dependence on glucose and glycolysis to support their maturation, activation-associated cytokine production, and T-cell stimulatory capacity. We have previously shown that this increase in glucose metabolism can be initiated by both Toll-like receptor (TLR) and C-type lectin receptor (CLR) agonists. In addition, we have shown that the TLR-dependent demand for glucose is partially satisfied by intracellular glycogen stores. However, the role of glycogen metabolism in supporting CLR-dependent DC glycolytic demand has not been formally demonstrated. In this work, we have shown that DCs activated with fungal-associated β-glucan ligands exhibit acute glycolysis induction that is dependent on glycogen metabolism. Furthermore, glycogen metabolism supports DC maturation, inflammatory cytokine production, and priming of the nucleotide-binding domain, leucine-rich-containing family, pyrin domain-containing-3 (NLRP3) inflammasome in response to both TLR- and CLR-mediated activation. These data support a model in which different classes of innate immune receptors functionally converge in their requirement for glycogen-dependent glycolysis to metabolically support early DC activation. These studies provide new insight into how DC immune effector function is metabolically regulated in response to diverse inflammatory stimuli.

## 1. Introduction

Dendritic cells (DCs) of the immune system are sentinel cells that can initiate both innate and adaptive immune responses [1,2]. By expressing an array of innate immune receptors including Toll-like receptors (TLRs), C-type lectin receptors (CLRs), and mannose receptors [3,4], DCs respond rapidly to a diverse array of pathogen-associated stimuli. Upon their activation, DCs upregulate the expression of proteins associated with their immune effector function, including the surface expression of costimulatory molecules and the secretion of immunomodulatory cytokines and chemokines. TLR-mediated DC activation induces changes in cellular metabolism signified by a rapid increase in glucose catabolism. This phenotype has been previously termed “glycolytic reprogramming,” and supports both the immune function and post-activation survival of DCs by satisfying the intense metabolic requirement associated with the rapid protein synthesis that occurs during DC activation [5,6,7,8]. Inhibition of aerobic glycolysis attenuates DC maturation and causes diminution of a variety of DC effector responses, from inflammatory cytokine secretion to T-cell stimulatory capacity [7,8,9].

We, and others, have recently shown that acute glycolytic reprogramming in DCs occurs in response to a variety of both TLR and CLR agonists [6,10,11]. We have further shown that CLRs require the signaling adapter molecule spleen tyrosine kinase (Syk) to induce glycolysis in DCs, whereas Syk signaling is dispensable for TLR-dependent glycolytic reprogramming [11]. In myeloid cells, phosphorylation of Syk induces a multifaceted inflammatory response including cytokine secretion [12], production of reactive oxygen and nitrogen species, and nucleotide-binding domain, leucine-rich-containing family, pyrin domain-containing-3 (NLRP3) inflammasome activation [13,14]. These studies suggest that the requirement for rapid glucose catabolism to support DC immune function is a centrally conserved cellular feature that spans multiple pattern recognition receptor families and upstream signaling pathways. This is notable because CLR-specific responses are nonredundant with TLR-mediated innate immune responses in certain contexts. For example, mice deficient for the CLR Dectin-1 display impaired immune responses against *Candida albicans*, resulting in compromised resistance to fungal infection [15].

Initial studies in the field of myeloid cell immunometabolism proposed that immune cells largely support activation-associated glycolytic metabolism by upregulating expression of glucose transporters to increase glucose uptake [16,17,18,19]. To this point, the inducible glucose transporter, GLUT1, has been well characterized as an important contributor to the long-term glycolytic reprogramming associated with myeloid cell immune responses [20,21]. Recently, we have proposed a more nuanced model of DC glucose homeostasis, whereby early glycolytic reprogramming in DCs is primarily driven by the catabolism of intracellular glycogen reserves, which preferentially support glucose-dependent citrate production as a signature metabolic feature of DC activation [22].

In the present study, we were tested DC dependence on intracellular glycogen to fuel early glycolysis induction and whether this dependence is conserved in both TLR and Syk-dependent CLR agonists. We hypothesized that this would be probable, given the conserved dependence on high rates of glycolysis for both categories of stimuli, but whether Syk-specific signals similarly regulate glycogen catabolism has not been previously demonstrated. In addition, although the importance of glycolytic reprogramming for DC effector function has been extensively studied in TLR-driven activation, there are only a few studies describing DC metabolic regulation in response to fungal-associated stimuli [10,11].

In this study, we have shown that glycogen-dependent glycolysis induction is similarly regulated in response to both TLR and Syk-dependent CLR agonists. Inhibition of glycogen phosphorylase (PYG), the rate-limiting enzyme in glycogen catabolism, attenuates TLR and Syk-dependent CLR-driven glycolytic burst, DC maturation, and inflammatory cytokine production. In addition, using NLRP3 inflammasome priming as a signature hallmark of early inflammatory activation, we have shown that glycogen metabolism plays an important role in regulating interleukin 1β (IL-1β) and caspase-1 levels in response to both TLR and Syk-dependent CLR agonists. Collectively, our study suggests that DCs rely on intracellular glycogen stores to fuel early glycolytic responses to both TLR and Syk-dependent CLR agonists. These data highlight the functional redundancy of the metabolic changes required for DC activation, adding evidence that CLRs, in addition to TLRs, drive early glycolytic reprogramming in DCs in a glycogen-dependent manner, and that this activity is important for sustaining the early inflammatory responses to both bacterial and fungal-associated ligands.

## 2. Materials and Methods

### 2.1. Mice and Reagents

C57BL/6J were purchased from Jackson Laboratory and bred in-house and maintained at the University of Vermont animal care facility under protocols (IACUC # 18-027, approved on 19 August 2019) approved by the Institutional Animal Care and Use Committee. Endotoxin-free lipopolysaccharide (LPS, *Escherichia coli* serotype O), Pam2Csk4 (Pam), Zymosan (Zy), Zymosan depleted (ZD), and nigericin were purchased from InvivoGen (San Diego, CA). Glycogen phosphorylase inhibitor (PYGib) CP-91149 was purchased from Selleckchem (Houston, TX). Antibodies for flow cytometry: 7-Aminoactinomycin D (7-AAD), anti-CD11c (clone N418), anti-CD86 (clone GL-1), and anti-CD40 (clone 3/23) antibodies, were purchased from BD Biosciences (San Jose, CA) and BioLegend (San Diego, CA). For Western blot analysis, cleaved caspase-1 (clone E2G2I) and cleaved IL-1β (clone E7V2A) antibodies were from Cell Signaling, and β-actin (clone 643,802) was purchased from BioLegend.

### 2.2. Mouse DC Culture and Activation

Bone marrow-derived DCs (BMDCs) were generated as described by Lutz et al. [23]. Briefly, bone marrow hematopoietic cells were differentiated in GM-CSF (20 ng/uL; Shenandoah Biotechnology Inc., Warwick, PA) in complete DC medium (CDCM), comprising RPMI1640, 10% FCS, 2 mM L-glutamine, 1 IU/mL Pen-Strep, and 55 μM beta-mercaptoethanol, for 7 days. On day 7, DCs were washed in CDCM and cultured at 2 × 10^5^ cells per 200 μL of media. For intracellular cytokine staining, cells were activated for a total of 6 h with an addition of GolgiPlug (BD Biosciences) after the first hour of stimulation.

### 2.3. Western Blot Analysis

DCs were lysed using lysis buffer with Pierce Protease and Phosphatase Inhibitors (ThermoFisher, Grand Island, NY). For cell lysate analysis, protein levels were quantified using the Pierce BCA Assay kit and normalized to 20–30 μg of total protein (depending on the individual blot) prior to running on 12% SDS-PAGE gels and subsequent transfer to nitrocellulose membranes. For cell supernatant analysis, 2–4 × 10^6^ cells were stimulated in 2 mL of media, and supernatants were concentrated 10-fold using StrataClean Resin (Agilent, Santa Clara, CA) to non-specifically concentrate all proteins in the supernatant. Cleaved caspase-1 and cleaved IL-1β blots were performed on these concentrated supernatant preparations.

### 2.4. Metabolism Assays

Extracellular acidification rate (ECAR) and oxygen consumption rate (OCR) were measured using Metabolic Flux Analyzer (Agilent/Seahorse Bioscience, 96-XF*^e^*).

### 2.5. Flow Cytometry and Cytokine Measurements

Abovementioned antibodies were used for flow cytometry. For intracellular staining of TNFα and IL-12 (BioLegend), cells were fixed in 4% paraformaldehyde, permeabilized in 0.2% saponin, and stained with the antibodies in flow cytometry buffer (PBS, 1% FBS). Samples were analyzed on a BD LSR II flow cytometer (BD Biosciences). For cytokine levels, supernatants were collected at indicated time points and measured by DuoSet ELISA kits (R&D Systems, Minneapolis, MN).

### 2.6. Statistical Analysis

Data were analyzed with GraphPad Prism software (version 8.4.0). Samples were analyzed using two-way ANOVA. ANOVA tests were post-calculated by Tukey’s multiple comparison test. Results are means +/− standard deviation, and statistical values are represented with an asterisk as significant when *p* values are equal to or below 0.05.

## 3. Results

### 3.1. Glycogen Metabolism Contributes to Glycolytic Metabolic Reprogramming in Response to Both TLR and CLR Agonists

To identify the role of glycogen metabolism in TLR and Syk-dependent CLR-mediated acute metabolic reprogramming in DCs, we used a panel of ligands specific to TLRs alone (lipopolysaccharide, LPS; Pam2Csk4, Pam), Dectin-1/2 alone (Zymosan depleted, ZD), or ligands that interact with both simultaneously (Zymosan, Zy), as we have previously published [11]. In these previously published studies, we showed that the Dectin-1/2 agonist, ZD, mediates glycolysis induction, DC maturation, and NLRP3 inflammasome priming in an entirely Syk-dependent manner, which allows us to isolate Syk-dependent signaling from other pathways employed either exclusively or coordinately with the other agonists in our experimental panel [11]. We first characterized the ability of these different agonists to induce acute glycolytic reprogramming in DCs by metabolic extracellular flux analysis (Agilent/Seahorse Biosciences). Cells were stimulated with the indicated agonists followed by addition of glycogen phosphorylase inhibitor (PYGib) to sequentially assess the level of glycolytic reprogramming in response to each agonist and the contribution of glycogen metabolism to glycolysis induction in real time (Figure 1). As we have previously published [11], all ligands tested induced a significant increase in rates of glycolysis compared to basal levels (Figure 1). Consistent with earlier work from our laboratory [22], LPS-dependent glycolytic reprograming was significantly attenuated by PYGib treatment (Figure 1A), as is the case for TLR2 agonist Pam (Figure 1B), TLR2/Dectin-1/2 dual agonist Zy (Figure 1C), and Dectin-1/2 agonist ZD (Figure 1D). These data indicate that glycogen metabolism significantly contributes to the early utilization of glucose in response to the ligands tested (Figure 1E).

### 3.2. Glycogen Metabolism Supports DC Maturation in Response to Both TLR and Syk-Dependent CLR Agonists

To test the role of glycogen metabolism in regulating DC maturation in response to both TLR and Syk-dependent CLR agonists, DCs were matured for 18 h with our panel of agonists in the presence or absence of PYG inhibitor (Figure 2). Consistent with previously published data [22], glycogen phosphorylase inhibition attenuated LPS-mediated upregulation of costimulatory molecules (Figure 2). In addition, we saw an impaired upregulation of costimulatory molecule expression with PYG inhibition in response to ligands that engage Dectin-1/2, but not with the TLR2-specific agonist (Figure 2). These data demonstrate a dependence on glycogen metabolism to support costimulatory maturation mediated by TLR4 and Dectin-1/2, but not by TLR2.

### 3.3. Glycogen Phosphorylase Inhibition Causes Deficits in Inflammatory Cytokine Expression and Nitric Oxide Production in Response to TLR and Syk-Dependent CLR Agonists

We and others have previously published that glucose and glycogen metabolism both contribute to LPS-mediated inflammatory cytokine production in addition to DC maturation-associated costimulatory molecule expression [6,22]. We have also shown that production of the potent microbicidal compound, nitric oxide (NO), is metabolically regulated in DCs [7,9]. To test the contribution of glycogen metabolism to early inflammatory cytokine responses, DCs were stimulated for 4 h with our panel of agonists in the presence or absence of PYG inhibitor and stained for intracellular expression of IL-12 (Figure 3A) and TNFα (Figure 3B). Consistent with the DC maturation data in Figure 2, PYG inhibition attenuated the induction of IL-12 production in response to LPS, Zy, and ZD ligands, but not Pam (Figure 3A). This effect was observed for both the percent of IL-12^+^ DCs induced and the average expression of IL-12 in IL-12^+^ cells. For TNFα expression, PYG inhibition did not alter the percent of TNFα^+^ cells for any agonist, but did attenuate the mean TNFα expression in TNFα^+^ cells for all stimuli (Figure 3B). To assess the contribution of glycogen metabolism to NO production, cells were stimulated for 18 h with our panel of agonists in the presence or absence of PYG inhibitor and nitrite accumulation in the cell culture media was measured. PYG inhibition significantly inhibited agonist-dependent nitrite accumulation for all stimuli tested (Figure 3C). Of note, Zy and ZD stimulation showed the highest sensitivity to PYG inhibition for all cytokine and NO measures, suggesting that Syk-dependent CLR activation may be more highly dependent on glycogen metabolism than what we have previously reported for TLR4 activation [22].

### 3.4. Glycogen Phosphorylase Inhibition Causes Deficits in IL-1β Secretion and NLRP3 Inflammasome Priming in Response to TLR and Syk-Dependent CLR Agonists

We next sought to test the contribution of glycogen metabolism to the priming of the NLRP3 inflammasome as indicated by the synthesis, release, and cleavage of caspase-1 and IL-1β. The NLRP3 inflammasome and IL-1β production are important inflammatory metrics for a number of reasons: first, IL-1β is produced early during myeloid cell activation and acts as a potent inflammatory amplifier, which makes it a notable biomarker of early activation; second, IL-1β secretion does not rely on the standard ER/Golgi secretory pathway, and thus potentially bypasses the glucose-to-citrate-to-fatty-acid-mediated ER enrichment that we and others have argued is an important mechanism of glucose-dependent cytokine regulation [6,22]; and third, we have recently published the dependence of NLRP3 inflammasome priming on glycolytic metabolism in DCs [11], but the role of glycogen-dependent metabolism in this process has not yet been defined. To test the role of glycogen metabolism in regulating early IL-1β responses, DCs were stimulated with our panel of agonists for 6 h in the presence or absence of PYG inhibitor and cell supernatants were collected for IL-1β detection by ELISA. PYG inhibition lessened IL-1β release significantly for all stimuli (Figure 4A). Notably, the ELISA used does not distinguish between uncleaved and cleaved IL-1β. To directly test the role of glycogen metabolism in supporting the generation of cleaved IL-1β, as a readout of NLRP3 inflammasome priming, DCs were stimulated with LPS for 5 h in the presence or absence of different doses of PYGib, with the NLRP3 inflammasome activator nigericin added during the final hour of stimulation. Cell lysates and supernatants were analyzed by Western blot for the generation of cleaved IL-1β and cleaved caspase-1. PYG inhibition suppressed LPS-driven generation of cleaved IL-1β and cleaved caspase-1 both intracellularly and extracellularly in a dose-dependent manner (Figure 4B). In combination with the data in Figure 3, these findings show that glycogen metabolism contributes to synthesis and secretion of early inflammatory cytokines regardless of the dependence on the ER/Golgi secretory pathway of the cytokines in question.

## 4. Discussion

TLR stimulation drives DCs to undergo glycolytic reprogramming in order to meet cellular anabolic demands associated with activation [8,9]. Blocking glycolysis in TLR-activated DCs impairs their ability to produce inflammatory cytokines and stimulate T cells [8,9]. TLR-driven early glycolytic reprogramming in DCs is mediated by Akt-dependent TBK1/IKKε signaling in response to both TLR and Syk-dependent CLR agonists [6,11]. We have proposed a model whereby early glycolysis induction in TLR-stimulated DCs is supported by rapid catabolism of intracellular glycogen [22]. Inhibition of the rate-limiting enzyme in glycogen catabolism, glycogen phosphorylase (PYG), attenuates inflammatory cytokine production, costimulatory molecule upregulation, and T-cell stimulatory capacity in DCs [22]. Although we have recently reported that CLR-specific activation of DCs induces glycolysis-dependent IL-1β production via the Syk-mediated signal transduction pathway [11], the contribution of glycogen metabolism to this pathway has not been previously defined.

In the present study, we used purified ligands to interrogate the contribution of specific receptor families to pathogen-associated innate immune agonists. In a physiological context, TLR and CLR receptors are often simultaneously engaged by innate immune cells, nevertheless, deficiencies in CLR molecules alone can lead to increased infection susceptibility, demonstrating that these pathways are not entirely redundant [24,25,26]. This study further demonstrates an evolutionary convergence between the metabolic regulation of TLR- and CLR-mediated innate immune activation, whereby acute glycolysis induction is a requirement for proper downstream effector functions mediated by both families of receptors. Futhermore, it reinforces the model that early glycolytic metabolism is fueled by the catabolism of intracellular glycogen reserves regardless of which innate immune receptor family is engaged.

One of the interesting outcomes of this work was the observation that PYG inhibition affected the different agonists tested to varying degrees. For instance, the dependence on glycogen metabolism for purified TLR2 agonist was generally blunted compared to other ligands tested. Other reports have proposed qualitative differences in cellular metabolic responses to distinct types of microbial-associated ligands. Human monocytes activated with either Pam3Csk4 or LPS display distinct metabolic responses, in which Pam3Csk4 induces increases in both glycolysis and oxidative phosphorylation (OXPHOS), whereas LPS preferentially supports aerobic glycolysis but not OXPHOS [27]. In addition, there is emerging evidence that different metabolic process may drive distinct immune effector functions. For example, in TLR-stimulated human monocytes, OXPHOS metabolism primarily supports phagocytosis whereas glycolysis preferentially supports inflammatory cytokine production [27]. We have previously shown that metabolic “compartmentalization” occurs even within the glycolysis pathway itself, whereby glycogen-derived carbons preferentially support TCA cycle-dependent citrate production as opposed to glucose-derived carbons [22], providing an intracellular precedent for metabolically linked functional compartmentalization described above.

The present study determined that both TLR and Syk-dependent CLR agonists induced acute glycolytic reprogramming in DC in a glycogen-dependent manner. Since cellular metabolic networks are inherently complex and exhibit multiple levels of redundancy and compensation, additional work at both the cellular and organismal levels is required to better understand how glycogen metabolism comprehensively controls DC activation. With a rising interest in metabolic regulation and consequent immune outcomes from various pathogenic stimuli, our work adds to an increasingly nuanced understanding of how glucose metabolism is regulated by DCs in response to different microbial stimuli. Understanding the complexity of metabolic regulation of innate immune cell responses to different pathogens remains an important issue in the field that warrants continued investigation.

## Figures and Tables

**Figure 1 cells-09-00715-f001:**
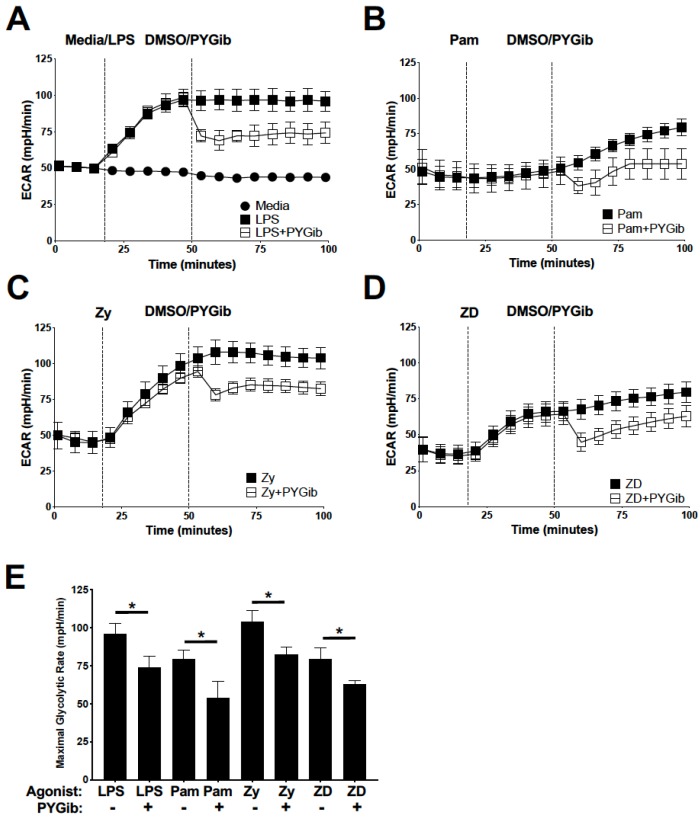
Glycogen metabolism contributes to glycolytic metabolic reprogramming in response to both TLR and CLR agonists: (**A**–**D**) extracellular flux analysis for real-time glycolysis rates were measured for DCs stimulated with the indicated agonists (LPS = lipopolysaccharide (**A**), Pam = Pam2CSK4 (**B**), Zy = Zymosan (**C**), and ZD = Zymosan depleted (**D**)), followed by subsequent injection of PYG inhibitor (PYGib). (**E**) Statistical comparison of maximal glycolysis rates following PYGib/DMSO control addition for each agonist. For all graphs, statistical values are represented with an asterisk (*) as significant when *p* values are equal to or below 0.05, with mean +/− SD shown, *n* = 4.

**Figure 2 cells-09-00715-f002:**
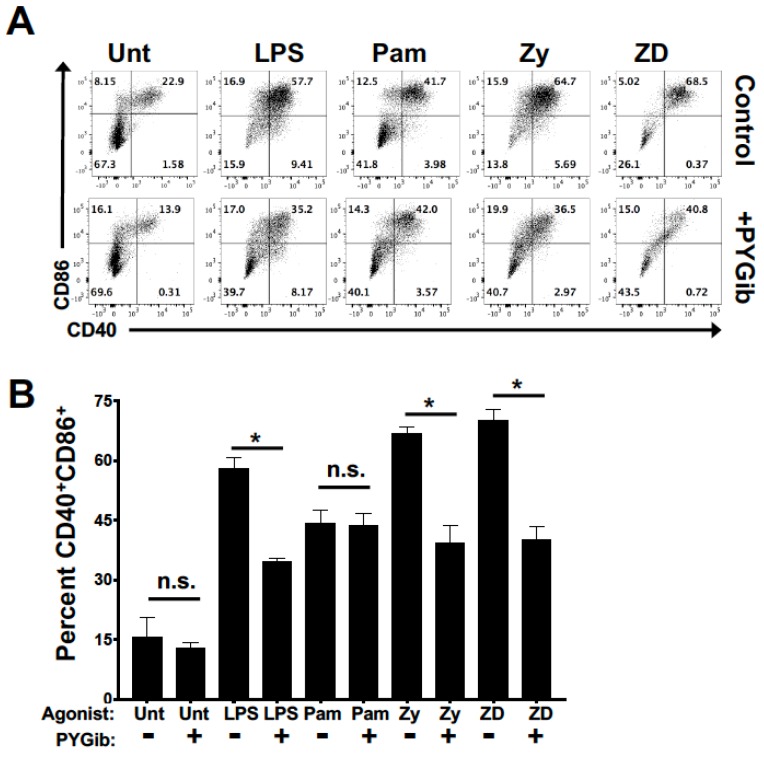
Glycogen metabolism supports DC maturation in response to both TLR and Syk-dependent CLR agonists: (**A**) DCs were stimulated for 18 h with the indicated ligands (LPS = lipopolysaccharide, Pam = Pam2CSK4, Zy = Zymosan, and ZD = Zymosan depleted) in the presence or absence of PYG inhibitor (PYGib) and then assessed by flow cytometry for maturation by CD86 and CD40 surface expression; (**B**) Percent of CD40 and CD86 double-positive cells for each condition as in (**A**). For all graphs, statistical values are represented with an asterisk (*) as significant when *p* values are equal to or below 0.05, with mean +/− SD shown, *n* = 4.

**Figure 3 cells-09-00715-f003:**
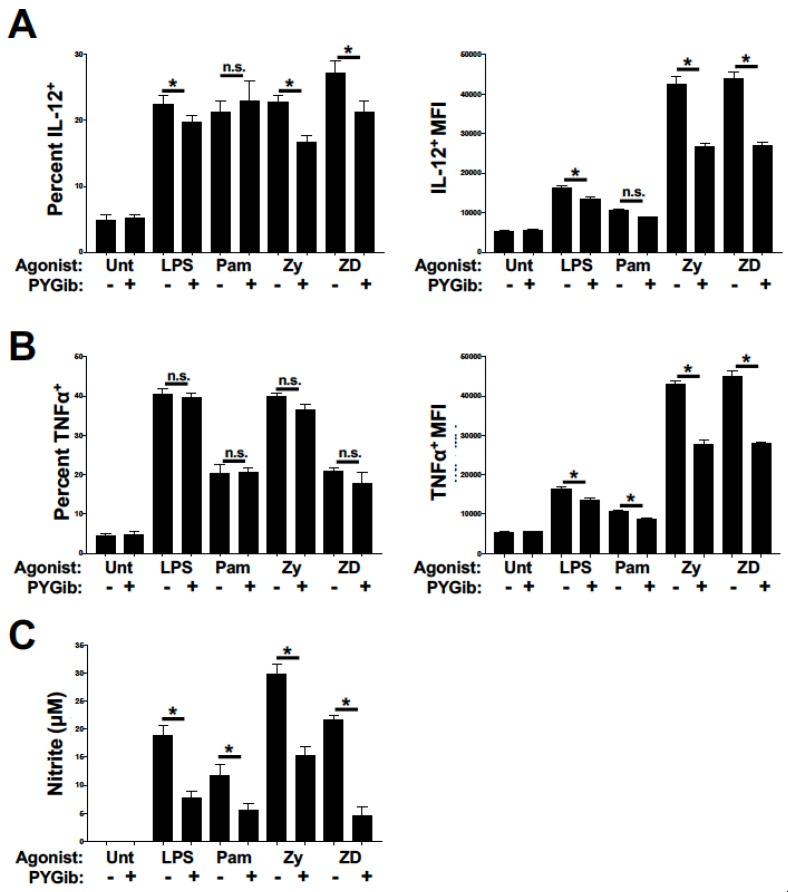
Glycogen phosphorylase inhibition causes deficits in inflammatory cytokine expression and nitric oxide production in response to TLR and Syk-dependent CLR agonists: (**A** and **B**) DCs were stimulated for 6 h with the indicated ligands and GolgiPlug reagent (LPS = lipopolysaccharide, Pam = Pam2CSK4, Zy = Zymosan, and ZD = Zymosan depleted) in the presence or absence of PYG inhibitor (PYGib), and then assessed by flow cytometry for intracellular expression of IL-12p70 (**A**) and TNFα (**B**). The percent of cytokine-positive cells (from total CD11c+ cells) is depicted in the left panels, while the mean fluorescence intensities of cytokine-positive cells are depicted in the right panels. (**B**) Percent of CD40 and CD86 double-positive cells for each condition as depicted in (**A**). (**C**) DCs were stimulated for 18 h with the indicated conditions as above, and supernatants were then collected and analyzed for nitrite accumulation by Griess assay. For all graphs, statistical values are represented with an asterisk (*) as significant when *p* values are equal to or below 0.05, with mean +/− SD shown, *n* = 4.

**Figure 4 cells-09-00715-f004:**
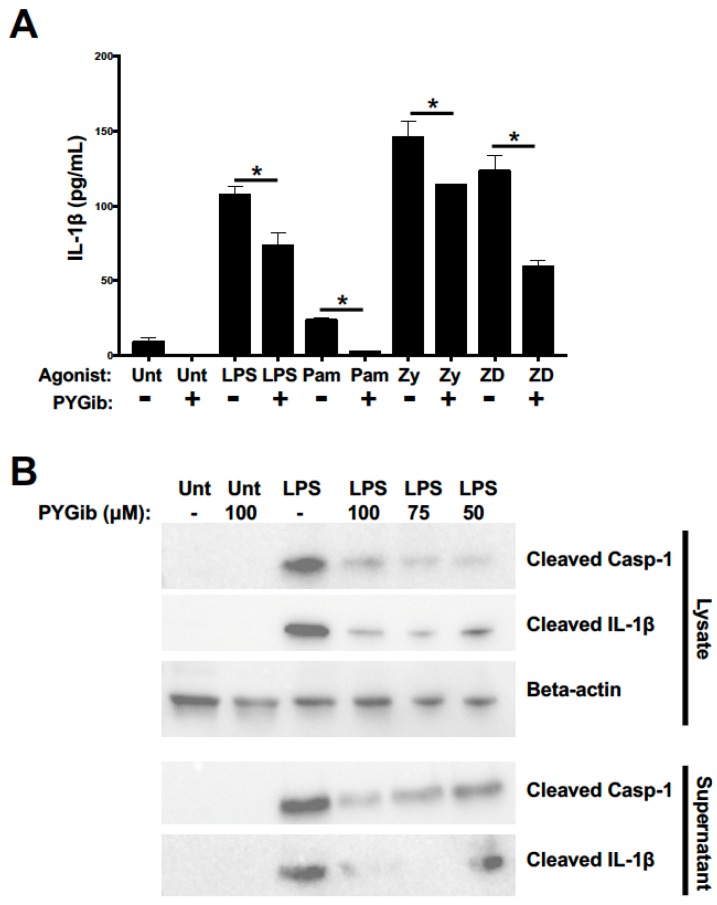
Glycogen phosphorylase inhibition causes deficits in IL-1β secretion and NLRP3 inflammasome priming in response to TLR and Syk-dependent CLR agonists: (**A**) DCs were stimulated for 6 h with the indicated conditions above and supernatants were then collected and analyzed for total IL-1β secretion by ELISA. For all graphs, statistical values are represented with an asterisk (*) as significant when *p* values are equal to or below 0.05, with mean +/− SD shown, *n* = 4. (**B**) DCs were stimulated for 5 h with the conditions indicated above with the last hour containing NLRP3 inflammasome activator nigericin. Cell lysates and supernatants were collected and analyzed for cleaved caspase-1 and cleaved IL-1β expression. Blots are representative of 3 independent experiments.

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
