# Peer review of "Glycogen Metabolism Supports Early Glycolytic Reprogramming and Activation in Dendritic Cells in Response to Both TLR and Syk-Dependent CLR Agonists"

_cells, 2020, doi:10.3390/cells9030715_

Round 1

Reviewer 1 Report

The authors explored the role of glycogen metabolism in dendritic cells (DC). They observed that activated DC demonstrate glycolytic induction partially dependent on glycogen metabolism. Using pharmacological approaches, the authors showed that glycogen metabolism play a role in DC maturation, cytokine production and priming of the NRLP3 inflammasome in response to TLR and CLR agonists.

The presented study is a valuable contribution to understand the metabolism of DC. The study matches with the journal scope. However, the following concerns should be addressed by the authors:

Major concern:

The weakness of this study is the fact that all experiments rely on the use of only one pharmacological drug, the glycogen phosphorylase inhibitor CP-91149, used at one concentration. In these conditions, we cannot exclude side-effects (off-target effects). The authors should perform additional experiments to demonstrate the real effects of these drug on its target and on glycogen metabolism. A dose-response should be also informative. Besides, it should be interesting to complete the pharmacological inhibition using another approach to inhibit PYG such as siRNA.

Other concerns:

In Figure 1A-D, the legend describing the Seahorse experiments are of bad quality and similar labeling (black vs white) should homogeneous in all panels. The authors should correct this.  It is unclear whether statistical analysis used SD or SEM as mentioned in Materials and methods. SD and SEM are not the same.

3.Figure 4 A: the level of IL1beta secretion are very low. Is it physiologically relevant ?

Author Response

Major concern:

The weakness of this study is the fact that all experiments rely on the use of only one pharmacological drug, the glycogen phosphorylase inhibitor CP-91149, used at one concentration. In these conditions, we cannot exclude side-effects (off-target effects). The authors should perform additional experiments to demonstrate the real effects of these drug on its target and on glycogen metabolism. A dose-response should be also informative. Besides, it should be interesting to complete the pharmacological inhibition using another approach to inhibit PYG such as siRNA.

Thank you for your thoughtful review of our manuscript. In previous studies (Thwe et al., Cell Metabolism, 2017). We have carefully vetted dose and specificity of the PYGib inhibitor, including parallel studies in mouse and human DCs and siRNA validation as well. In addition, the drug dose-response in Figure 4 confirms are previous work in terms of the narrow efficacy window of the drug (between 50-100 uM). This is consistent with the relative pharmacological drug-resistance that we routinely see in DCs (compared to lymphocytes for example).

Other concerns:

In Figure 1A-D, the legend describing the Seahorse experiments are of bad quality and similar labeling (black vs white) should homogeneous in all panels. The authors should correct this.  It is unclear whether statistical analysis used SD or SEM as mentioned in Materials and methods. SD and SEM are not the same.

We have corrected for clarity and consistency the figure labels, legends, increased the font size on the axis labels, and clarified which statistical variance (SD) was used throughout the manuscript.

3.Figure 4 A: the level of IL1beta secretion are very low. Is it physiologically relevant ?

The IL1beta levels indicated the very early (6 hour) production of this response, which is when we have shown glycogen metabolism to matter the most over the course of DC activation. Nevertheless, we see significantly more IL1beta production at 24 hours (data not shown in the manuscript), with the relative trends being preserved and remaining statistically significant.

Reviewer 2 Report

The manuscript entitled “Glycogen metabolism supports early glycolytic reprogramming and activation in dendritic cells to both TLR and Syk-dependent CLR agonists” by Kylie D. Curtis et al. is a continuation of research over glycolytic reprogramming in DCs and the role of TLR signalling in promotion of aerobic glycolysis. The authors extend their research on metabolic regulation of DC immune function in response to different inflammatory stimuli and proposed interesting view how metabolism influences DC activation.

Hence, I am certain that manuscript is acceptable in this form for Cells.

Author Response

We would like to thank the reviewer for their reading and evaluation of the manuscript, and for the favorable evaluation of our work.

Round 2

Reviewer 1 Report

The authors have addressed most of the concerns regarding the manuscript.